# Effects of Positive Reinforcement Training and Novel Object Exposure on Salivary Cortisol Levels under Consideration of Individual Variation in Captive African Elephants (*Loxodonta africana*)

**DOI:** 10.3390/ani11123525

**Published:** 2021-12-10

**Authors:** Susan Hambrecht, Ann-Kathrin Oerke, Michael Heistermann, Johannes Hartig, Paul W. Dierkes

**Affiliations:** 1Magdeburg Zoo, Zooallee 1, 39124 Magdeburg, Germany; 2Bioscience Education and Zoo Biology, Goethe-University Frankfurt, Max-von-Laue-Straße 13, 60438 Frankfurt am Main, Germany; dierkes@bio.uni-frankfurt.de; 3Endocrinology Laboratory, German Primate Centre, Leibniz Institute for Primate Research, Kellnerweg 4, 37077 Göttingen, Germany; akoerke@dpz.eu (A.-K.O.); MHeistermann@dpz.eu (M.H.); 4Educational Measurement, Teacher and Teaching Quality, Leibniz-Institute for Research and Information in Education, Rostocker Straße 6, 60323 Frankfurt am Main, Germany; hartig@dipf.de

**Keywords:** zoo elephants, individuality, physiological stress, glucocorticoid response

## Abstract

**Simple Summary:**

Regular training for medical or enrichment purposes and the provision of environmental enrichment, such as varied feeding schedules and novel objects, are part of the management of African elephants in zoos. The present study aimed to find out whether training and enrichment in the form of a novel object induced physiological changes in captive African elephants. We repeatedly sampled the saliva of ten animals (three zoos) before and after training and the exposure to a novel object for the analysis of cortisol and as a measure of stress and arousal. We found high salivary cortisol levels before and low levels after training. A novel object, in contrast, moderately increased the salivary cortisol levels. Moreover, males and young elephants showed lower salivary cortisol levels than females and old elephants, respectively. The zoo, handling method (free vs. protected contact to keepers), reproductive and social status, however, did not influence the salivary cortisol levels of the animals studied. We conclude that the relatively high cortisol values before training could be due to anticipation of the training event. A novel object, in contrast, may have evoked arousal, which led to the observed cortisol increase. In addition, understanding why animals differ in stress responses will help to manage stress in zoo elephants with the goal of ensuring their welfare.

**Abstract:**

Dealing with potential stress in species that have high husbandry requirements, such as elephants, is a challenge for zoos. The objective of the present study was to determine whether positive reinforcement training (PRT) and exposure to a novel object (NOV) for enrichment induced a salivary cortisol response indicative of activation of the hypothalamic–pituitary–adrenal (HPA) axis and which factors determine individual variation in this regard in captive African elephants. We repeatedly sampled the saliva of ten animals (three zoos) for the analysis of cortisol (SACort) before and up to 60 min (in 10–15 min intervals) after the onset of PRT (three repeats) or NOV (nine repeats), which lasted 10 min. There was considerable individual variation in SACort in response to PRT or NOV. Using mixed models, we were able to control these and to reveal that PRT was associated with high SACort before and relatively low SACort after PRT, while NOV induced a moderate SACort increase. The individual differences in SACort were related to age and sex (NOV), while the effects of zoo, handling method (free vs. protected contact) and reproductive and social status were variable. We conclude that positive affective states, such as anticipation or arousal, should be taken into account when interpreting the differences in the SACort responses between PRT and NOV. In addition, understanding the individuality of stress will support management decisions aimed at promoting captive elephant welfare.

## 1. Introduction

### 1.1. Glucocorticoid Levels as an Indicator of Stress

Stress is a complex state of the organism that occurs when the individual perceives an external or internal stimulus as a threat to homeostasis [1]. The perception of such a stressor elicits a coordinated stress response that consists of behavioral and physiological changes. These are vital because they restore homeostasis, i.e., enable stress coping. The stress response, or more precisely, its impact on the organism, defines whether the stress to which the organism is exposed is positive or negative [2]. An adequate stress response restores homeostasis within a short time and at low biological cost, and is, therefore, adaptive. In such a case, stress can be considered positive, as it does not affect well-being [2] and may even contribute to the stress resilience of the organism [3]. Inadequate, excessive and/or prolonged stress responses, however, that are caused by long-term (i.e., chronic) and/or severe stressors can significantly alter biological functions, adversely affecting the welfare of the organism. In this case, the stress is negative and is referred to as distress [2].

The physiological changes of the stress response include the increased activity of the sympathetic nervous system (SNS) [4,5] and increased glucocorticoid (GC; cortisol/corticosterone) secretion from the hypothalamus–pituitary–adrenal axis (HPA axis) [6]. These steroid hormones, in general, contribute to normal and stress-related biological functioning [7,8]. Furthermore, GC secretion is influenced by both negative (e.g., fear, anxiety and pain) and positive affective states (e.g., arousal, excitement, anticipation and reward) [8].

As GCs play a key role in the physiological stress response, their measurement is frequently used as an indicator of stress and welfare in wild and captive animals, including African elephants (*Loxodonta africana*). Different sampling matrices can be analyzed for GCs. Hormone analyses in excreta (urine, feces) provide integrated measurements of the amount of GCs released, metabolized and excreted over a certain period of time [9]. These so-called non-invasive methods have, therefore, only limited suitability for the assessment of short-term changes of GC levels caused by the diurnal variation of cortisol secretion (but see: [10,11,12]) or acute GC responses to specific stressors [13]. Blood and saliva allow for such a real-time assessment of GC levels. In contrast to urine and feces sampling, however, the sampling of blood and saliva is more challenging and partly invasive (blood) as both require a certain degree of interaction between the researcher and the animal during sampling, which bears the risk of inducing confounding GC elevations [14]. Therefore, the collection of blood or saliva from free-ranging animals is often not feasible but can be trained as part of daily routines in zoo animals. In the course of training, the animals become used to the method of sampling and no longer react with a stress response to the sampling procedure [15,16,17]. Many studies on stress in African elephants rely on the measurement of cortisol, the major GC in elephants, or its metabolites in excreta to study the effect of environmental conditions on physiological stress, both in the field [18,19,20] and in the zoo [21,22,23,24,25]. Feces, e.g., [26,27], and blood, e.g., [21,24], are the most frequently used sampling matrices for GC measurements in African zoo elephants. Saliva has also been used a few times in this species as a GC sampling matrix [28,29,30,31] and can be collected minimally invasively [32] from captive African elephants as part of routine training, as the mouth cavities of the animals are regularly inspected. The animals can, therefore, be easily habituated to saliva collection, which minimizes the abovementioned risk of causing GC elevations by sampling itself.

It has been shown in various species, including African elephants, that the salivary GC levels reliably reflect the GC levels in blood [29,33,34]. To the best of our knowledge, however, information on the blood–saliva time lag or the time course of both blood and salivary cortisol levels in response to acute stressors does not exist for any elephant species. In other mammals, salivary cortisol increased immediately from baseline and generally reached peak levels 10–50 min after the onset of a stressor (human: [35]; cattle: [36]; chimpanzee, *Pan troglodytes*: [37]; domestic dog: [38]; but see [39] for horses). Recovery to baseline levels is more variable and takes from 60 min (dog: [38]; horse: [40]) to several hours (chimpanzee: [37]).

We are aware of only one study in captive African elephants [31] in which several saliva samples were collected in 15-min intervals after different treatments. Salivary cortisol levels did not show a consistent pattern of increase or change in response to training and enrichment in the three animals involved in the experiment. In response to 15 min of isolation with a dominant conspecific, however, salivary cortisol levels were elevated 15 min after the end (30 min after the onset) of isolation and began to decline by 30 min after the end in all animals. These results suggest that in African elephants, peak salivary cortisol secretion following a stressor occurs with a similar time lag, as in other species (see above).

### 1.2. Individual Variation of Glucocorticoid Levels

Individuals differ in the GC stress response to the same stressor [41] with some showing a GC increase and others displaying an attenuated or no GC response at all [1]. This inter-individual variation was found independently of the sampling matrix (blood, feces, urine or saliva) and can have the following various causes, among others: age (chimpanzee: [42]), sex (dwarf hamster, *Phodopus campbelli*: [43]; domestic sheep: [44]), personality/coping style (reviewed in [45]), social environment including social status [46], or unknown causes (spotted hyena, *Crocuta crocuta*: [47]; beluga, *Delphinapterus leucas*: [48]; domestic dog: [49]). In addition, individuals may differ in their GC responses based on previous experiences (tiger, *Panthera tigris*: [50]; domestic dog: [51]). Greater experience due to repeated exposure to a stressor has been shown to lower GC responses due to the habituation of the HPA axis [52] and the perception of predictability and controllability in the animals [53]. Moreover, GC secretion in vertebrates follows an endogenous circadian rhythm [54], resulting in variation within and between individuals depending on the time of day at which GC levels are measured.

In both elephant species, studies revealed inter-individual variation in GC responses to captive husbandry situations in all sampling matrices (African elephant, blood: [21], saliva: [31]; Asian elephant, feces: [55], blood and urine: [56]). The individual factors shown to influence GC levels within or between individual elephants include age (African elephant: [57]; Asian elephant: [56]), sex (wild African elephant: [19]), facility [23], personality (African elephant: [29]; Asian elephant: [55]), day time [28,29,58,59,60] and reproductive status or ovarian cycle phase (African and Asian elephant: [61]; African elephant: [62,63]; Asian elephant: [56,64,65]).

In addition, interactions between different factors can be expected in determining GC levels, for example, between age and reproductive status, as recently shown in Asian elephants [56].

In general, the ubiquitous finding of individuality in the GC response underscores the need to assess stress at the individual level, taking into account factors that potentially explain inter- and intra-individual variation in stress perception and response.

### 1.3. The Management of Stress in Zoo Animals

The daily life of free-ranging animals is far from being stress-free, as it includes stressors such as hunting or being hunted, mating, establishing and maintaining social hierarchies, weather extremes, food scarcity, increasing anthropogenic disturbances, disease and injury [18,66,67,68,69]. Indeed, captive conspecifics do not face many of these natural stressors, but are instead or additionally confronted with a higher level of potential (di)stress due to the lack of physical and psychological stimulation, lack of choice and control and prevention of the performance of behavioral needs [69,70,71]. Environmental enrichment and positive reinforcement training are management tools aimed at counteracting these psychological stressors of zoo animals. They have been shown to reduce stress [15,72], enable animals to experience adaptive, i.e., beneficial, stress [73,74] or proactively prevent (di)stress from arising by promoting stress coping abilities and resilience [74,75,76,77,78].

Environmental enrichment (hereafter enrichment) includes a variety of activities, devices and stimuli that aim to increase the physical, temporal and social complexity of the environment for captive animals in order to provide those environmental stimuli that are necessary for optimal physical and psychological animal welfare [70,79]. There is a strong presumption that enrichment reduces long-term stress because it gives the animal more opportunities for behavioral responses to cope with the situation [70,80]. Enrichment also includes stimuli that excite animals in the short-term, for example, novel objects, which are a common form of enrichment [78,81,82] and are also recommended as a sensory stimulus for elephants [83]. Since new situations also occur in natural habitats [84], a novel object introduced to a captive-housed animal can be considered as a natural stressor. The reaction to a novel object is a result of the interaction of curiosity (i.e., the tendency to approach, called neophilia) and fear (i.e., the tendency to avoid, called neophobia) [85]. Fear is known to induce physiological stress [86]. Therefore, it is conceivable that novel objects have been found to trigger GC increases in cattle [86,87] and birds [88,89,90]. Thus, the impact of novel objects as enrichment needs to be evaluated and monitored to avoid the negative stress of excessive novelty. Neophobia, however, may decline over time, as the use of novel stimuli provide opportunities to learn to cope with novel situations, which increases resilience [91]. A novel object, therefore, may be a form of enrichment that allows short-term adaptive stress caused by positive stimulation.

Positive reinforcement training is the preferred form of training in zoos because it gives the animal choice and control over its environment by actively learning the contingency between its action and the reaction of the environment. On the other hand, the animal voluntarily cooperates in positive reinforcement training and can experiment with various behavioral responses without fearing negative consequences [92]. Positive reinforcement training is thus usually considered beneficial for the animals [93]. Thus far, studies examining the effect of routine positive reinforcement training on GC levels could not find any GC increases (African elephant: [31]; primates: [94,95]) suggesting that positive reinforcement training is not perceived as stressful by the animals involved. Compared to routine training, however, novel training can induce elevations in GC levels, as shown in the African elephant [31].

### 1.4. Rationale of the Present Study

Zoos face the challenge of designing a captive environment to meet the species-specific and individual needs of the animals in order to mitigate (di)stress and preserve animal welfare. This challenge is all the greater for species, such as elephants, that at the same time have high demands on their husbandry and whose welfare is in the public focus due to their popularity [96]. Positive reinforcement training and enrichment play an integral role in meeting this challenge [83,91,97]. The aim of the present study, therefore, was to examine the immediate physiological effect of these “stress management tools” as measured by salivary cortisol concentrations (hereafter: SACort) in captive African elephants. We predicted that (i) SACort should not rise in response to positive reinforcement training but should rise in response to novel object exposure (see Section 1.3) and, therefore, differ between positive reinforcement training and novel object exposure; (ii) the variation in SACort is related to age, sex, social status, reproductive status, zoo and/or handling method (free vs. protected contact) of the individual (see Section 1.2).

## 2. Materials and Methods

### 2.1. Animals and Management

We collected saliva samples for cortisol analysis from ten adult African elephants (*Loxodonta africana*), which were housed in the following German zoos: Opel-Zoo Kronberg, Grüner Zoo Wuppertal and Thüringer Zoopark Erfurt. In each zoo, animals were studied twice, with the two study periods separated by one year (2016, 2017). The study periods within each zoo took place in the same season (Kronberg: spring, Wuppertal: summer, Erfurt: fall/winter). Table 1 presents the information on the individual animals examined.

The elephant housing in all zoos was characterized by naturalistic outdoor enclosures and indoor areas with water basins, mud wallows, sandpits and scratching surfaces as well as several gated stalls (boxes) to separate individual animals. Depending on the weather conditions, elephants had access to outdoor enclosures. Females and their offspring were housed in stable social groups. Bulls had occasional access to the females. A new cow joined the Erfurt group in April 2017 and was fully integrated by the beginning of the study period in fall of that year.

### 2.2. Treatments

#### 2.2.1. Positive Reinforcement Training

In positive reinforcement training (abbreviated as PRT), the keeper reinforces desired behavioral responses to specific commands with pleasurable rewards [92]. In all three zoos, positive reinforcement training was part of the routine management of the elephants. The animals, therefore, knew the procedure of PRT, which included separation from the group in the training area, mostly a box (in the case of the cows, the group was always in sight to avoid separation stress), performance of PRT and release back into the group. This basic procedure applied to all three zoos. In addition, only known behavioral patterns were requested from the animals, such as lifting the foot, presenting the body side, presenting the ear and backing away from the keeper. To minimize the influence of the keeper (trainer) on the cortisol measurements, PRT was always performed by the same keepers whenever possible. Animals were trained alone except for SA and TI (mother and daughter), who were trained simultaneously to prevent social stress due to separation. PRT was conducted three times (trial 1–3) per animal in 2016 and was not carried out in 2017 after we realized in the year before that salivary cortisol was not affected.

All animals involved in the present study received positive reinforcement training regularly for medical and enrichment purposes. PRT varied according to the training level of each animal and the routine of PRT in the respective zoo. The latter was mainly related to the difference in handling methods. All elephants in Kronberg and Erfurt and the adult bull in Wuppertal were trained in protected contact, while the cows in Wuppertal were trained in free contact (Table 1). The primary reinforcer was always sought-after food (bread, fruit, vegetables or pellets). For the present study, however, the elephants were rewarded only with bread in order to minimize a potential influence of food on salivary cortisol [100,101,102]. A secondary reinforcer was only used in Kronberg and Erfurt in the form of a clicker. In addition, the elephants in Kronberg and Wuppertal were trained and showered at the same time. The keepers in Erfurt, however, did not shower the animals during PRT and also trained the elephants with the help of two target sticks, a tool that is frequently used in animal training. After an elephant has been conditioned to touch or follow the target stick, it can be guided inside the box.

#### 2.2.2. Novel Object Exposure

The animals were exposed singly to one novel object (exposure to novel object is abbreviated as NOV) each, except for the mother–daughter pair SA and TI, who were tested simultaneously (such as in PRT, see Section 2.2.1). They were offered two equal novel objects of the same type four to five meters apart in order to reduce competition for the novel object and allow both animals to be able to engage independently from each other with one novel object [103]. In all animals, the novel object was placed in the arena, a familiar part of the enclosure, not visible to the animal. The exposure to the novel object began when, as the door between the box and test arena opened, the animal looked in the direction of the novel object for the first time. The animal was able to enter and leave the arena voluntarily, since the door to the box remained open. Each animal was exposed to a novel object three times in 2016 (trial 1–3) and six times in 2017 (trial 4–9). Novel objects were considered as novel, as they were not used in routine management before. They were of different colors, materials and sizes (ball, traffic cone, white tarpaulin, blue barrel, air mattress, etc.). A different novel object was presented in each trial and each object was used only once. The same novel objects were presented to all animals in the same order. Unfortunately, we had to deviate from this rule in three trials at one zoo each, because it turned out that the respective novel object was either not robust enough or the elephants in the respective zoo had already been confronted with the respective object before.

### 2.3. Saliva Sample Collection for Cortisol Analysis

The stress potentially related to PRT and NOV examined in this study was measured only by cortisol in elephant saliva. A change in cortisol level indicated a physiological response to PRT or NOV. The use of this method in the present study did not imply any valuation of stress in terms of its effect on elephant welfare. The addition of measurement of other stress indicators to this method is addressed in the discussion (see Section 4.3).

We took saliva samples before (pre) and after (post) positive reinforcement training and the exposure to a novel object. In total, 589 samples were analyzed. Saliva was sampled by wiping out the mouth (cheek pouches, over and below the tongue) with a swab (Salivette^®^ Cortisol, Sarstedt, Nümbrecht, Germany), which was fixed in a clamp or stainless-steel tube. The saliva samples were frozen at −18 °C immediately following collection. The animals were habituated to the sampling procedure during regular training prior to the first study period and were sampled according to a standardized protocol, which included determination of social context (see Section 2.3.1), sampling schedule (see Section 2.3.2) and experimental set-up (see Section 2.3.3).

#### 2.3.1. Social Context

Per day and trial, a single animal was examined, except for SA and TI, who were studied simultaneously (see Section 2.2.1 and Section 2.2.2).

Since the separation of females from their group might trigger social stress, which, in turn, might confound with cortisol changes in response to the performed treatment (PRT or NOV), cows were never separated from their calves. In addition, all cows, except for one cow (SW) and her calf, had visual and limited tactile contact with the rest of the group throughout the saliva sampling period and during the treatment. SW received regular individual supplemental feeding, which required separation from the group. SW and her calf were thus accustomed to temporary separation from the group.

#### 2.3.2. Sampling Schedule

Sampling occurred at fixed daytimes to minimize the influence of the diurnal rhythm of cortisol secretion [30]. The time of day of sampling differed between zoos due to differences in management routine but was consistent within zoos during the whole study periods: 13:00 in Erfurt, 14:00 in Wuppertal, 18:00 in Kronberg. In addition, factors such as the keeper, food and location of sampling were held as constant as possible to promote consistency in the sampling context.

Only one trial of either PRT or NOV took place per day. Furthermore, each animal was examined only once per week (i.e., one trial of either PRT or NOV per animal and week) and the animals were consecutively exposed to one trial of the same situation per week in a stable order (e.g., trial 1 of animal A on Monday, trial 1 of animal B on Tuesday, etc.).

#### 2.3.3. Experimental Set-Up

The experimental set-up was the same for both PRT and NOV: In each trial, the animal was initially separated from the group and shifted into the box, a procedure to which the elephants were accustomed to as part of daily routines. Following the separation of the animal, the first saliva sample was collected as a control sample in order to obtain the salivary cortisol level before the elephant was subjected to the specific treatment (SACort0). The subsequent treatment (PRT or NOV) lasted 10 min until PRT was terminated, or for NOV, the animal was moved into the box and the door to the arena was closed. The first post-sample was collected immediately after termination of PRT or NOV (which was approximately 10–15 min after the pre-sample depending on the duration of moving the elephant to the sampling location). Afterwards, saliva samples were collected every 10 min (Kronberg) or 15 min (Wuppertal and Erfurt) for a total period of 60 min after onset of the treatment to examine the cortisol response to the treatment. The animals remained separated from the group in the box after PRT or NOV to facilitate saliva sample collection and to allow for the recovery of salivary cortisol levels in a constant environment. After the collection of the last sample, the elephant received a special reward (fruit, vegetables and concentrate feed) and was released back into the group.

The experiment was terminated if the animals were found to be excessively stressed during NOV, PRT, saliva sampling or the one-hour separation from the group.

Appendix A shows the number of saliva samples collected in each situation, animal and sampling time. The number of saliva samples collected per sampling time corresponded to the number of trials of the respective treatment (see Section 2.2.1 and Section 2.2.2). Before PRT and at each sampling time post-PRT three samples were collected per animal according to the three trails conducted. Three samples were taken per animal and sampling time in NOV in 2016 (3 trials) and six samples in 2017 (6 trials). In some cases, the sample size is smaller (Appendix A) due to saliva volume being too small for analysis (i.e., <50 µL) or shortening of the sampling period after PRT or NOV to the samples at 10 to 30 min.

### 2.4. Cortisol Analysis

We measured SACort with a microtiter plate enzyme immunoassay (EIA; [104]), which was used successfully for the assessment of physiological stress via SACort analysis in a number of other species [94,105]. We also biologically validated and successfully applied this assay previously to assess the diurnal rhythm of cortisol secretion in our study subjects [30]. All samples were measured in duplicate as described in detail in [30].

Samples with a coefficient of variation (CV) of >10% between duplicates, were re-measured. Assay sensitivity at 90% binding was 12 pg/mL. Inter-assay CVs, assessed by replicate determinations of high- and low-value quality controls run in each assay, were 7.5% (high) and 10.4% (low), and intra-assay CVs were 5.0% (high) and 6.8% (low). SACort is expressed as ng/mL.

### 2.5. Statistical Analysis

We used R (version 3.6.0, [106]) for data analysis. Significance level was *p* < 0.05. We also indicate when results approached significance (*p* < 0.1).

We applied linear mixed models (LMM) using the function lmer of the R-package lme4 (version 1.1-21; [107]) in order to examine the effect of specific parameters on SACort. LMMs controlled for non-independence of data based on the repeated saliva sampling in each animal by fitting trial nested in animal-ID as a random intercept in all LMMs. In other words, the LMM approach allowed us to separate the considerable intra- and inter-individual variation in SACort levels and time courses, as shown in Appendix A, from the variation caused by the parameters of interest.

These parameters were fitted as fixed effects and were either experimental (sampling time, situation and study period) or individual (age, sex, zoo, social and reproductive status (only in females) and handling method (only in females)) variables. We recognize that specific variables are related to each other, e.g., zoo and handling method, and therefore, using fewer variables would have been a more direct approach. However, following [68], we decided on a broader exploratory approach because of the unknown nature and complexity of relationships between SACort and influencing variables in captive African elephants. The LMMs are described in detail in the following sections.

In total, five LMMs were fitted to examine the two predictions detailed in Section 1.4. Table 2 gives an overview of the fixed effects included in each of these LMMs. Three LMMs used the data of all ten animals (“All animals/PRT and NOV”, “All animals/PRT” and “All animals/NOV”). The remaining two models (“Females/PRT” and “Females/NOV”) were for the data from the seven females only, as these LMMs were supposed to represent the influence of the individual parameters of social status and handling method, which were only recorded for the females, on SACort. In order to examine the treatment-specific cortisol response, we additionally set up LMMs for each treatment (“All animals/PRT”, “All animals/NOV”, “Females/PRT” and “Females/NOV”).

All models were fitted using the restricted maximum likelihood estimation (REML) and included the BOBYQA (Bound Optimization BY Quadratic Approximation; [108]) function as optimizer. The significance of fixed effects was tested using *t*-tests (Satterthwaite’s method) of the R package lmerTest (version 3.1-0, [109]). As we expected a cortisol increase relative to the intercept in response to NOV (see Section 1.4), the *t*-tests in the case of the fixed effects of the NOV models related to the sampling time were one-sided. The remaining *t*-tests were two-sided.

We log_10_-transformed salivary cortisol values (log_10_SACort) in order to fulfill the normality assumptions of the models (assessed using Q-Q-plots and histograms). For visualization of SACort responses we used the original non-transformed data.

#### 2.5.1. Effect of the Treatment on SACort

To test our first prediction of the effect of the treatment (PRT and NOV; see Section 1.4), the treatment, sampling time as a linear effect and, based on the expected trend of an acute cortisol response (see Section 1.1), a quadratic sampling time term defining 30 min as the peak of the parabola (${\left[\mathrm{sampling}\;\mathrm{time}-30\right]}^{2}$) were entered as fixed effects in the LMM “All animals/PRT and NOV” (Table 2). Sampling time was also fitted as a fixed effect in the remaining four LMMs. The study period (2016, 2017) was additionally considered as a fixed effect in the LMMs “All animals/PRT and NOV”, “All animals/NOV” and “Females/NOV” to control for the variation of SACort depending on the year in which NOV was performed (PRT was only conducted in 2016) (Table 2).

#### 2.5.2. Effect of Individual Parameters on SACort

To test the second prediction (see Section 1.4), age class, sex, zoo, social status (only in females) and handling method (only in females) were fitted as fixed effects in addition to the experimental factors (sampling time, treatment and study period) (Table 2). Age class was included in all five LMMs. Sex was fitted as a fixed effect in the LMMs referring to the data of all animals (“All animals/PRT and NOV”, “All animals/PRT” and “All animals/NOV”). Social status and handling method were only considered for females (LMMs: “Females/PRT” and “Females/NOV”).

The effect of reproductive status or ovarian cycle phase on female SACort was analyzed on an individual level. Separate LMMs were generated for each cow that had at least one change in reproductive status or ovarian cycle phase during the whole study period and exhibited ovarian cycles at some point during the study (Table 1). Reproductive status or ovarian cycle phase were fitted as a fixed effect and the trial as a random effect.

## 3. Results

### 3.1. Effect of the Treatment on SACort

The results confirm our first prediction that the SACort response differs on average between the two treatments positive reinforcement training (PRT) and exposure to a novel object (NOV). Figure 1 illustrates that the SACort in response to PRT was on average lower than the SACort in response to NOV. In addition, the average time courses of the SACort ran in opposite directions. The SACort after PRT decreased with the lowest average SACort level at 15 min being 23% lower than the average SACort before PRT at 0 min. The SACort after NOV, in contrast, increased. The average SACort increase was, however, small, with the highest SACort levels detected at 30 min post-NOV being 10% (NOV in 2016) and 20% (NOV in 2017) higher compared to the SACort levels measured prior to the onset of NOV at 0 min.

The results of the LMM “All animals/PRT and NOV” show that the treatment was a highly significant predictor of log_10_SACort. When controlling for the other predictors in the model (see Table 2), the log_10_SACort in PRT was 0.14 units lower than in NOV (*SE* = 0.02, *t* = −6.74, *p* ≤ 0.001, see Appendix A for detailed results of the LMM). The results of the treatment-specific LMMs (“All animals/PRT” and “All animals/NOV”) shown in Table 3 indicate that the quadratic term (and not the linear term) of the sampling time was a significant predictor of log_10_SACort in both treatments. The direction of the temporal cortisol change, however, differed between the treatments: the log_10_SACort change over time after PRT followed an upward opening parabola (initial decrease, minimum and increase compared to the intercept (log_10_SACort0 before PRT)). The SACort response to NOV showed the opposite trend: the log_10_SACort time course was a downward opening parabola (initial increase, maximum and decrease compared to log_10_SACort0). In addition, the LMM results show that the study period in which NOV occurred significantly influenced log_10_SACort (Table 3), with log_10_SACort in NOV being lower on average in 2017 than in 2016.

### 3.2. Effect of Individual Parameters on SACort

In line with our second prediction, the variation in SACort was related to specific individual factors. Figure 2 illustrates age class and sex differences in SACort in response to NOV in 2017 as an example. Male and relatively young elephants had a lower average SACort in response to NOV than female and relatively old elephants. For example, the SACort of the three males, TU, KB and TA, were lower on average than that of the females, except for the youngest females, CH and TI. Accordingly, the LMMs showed that age class and sex were significant predictors of the average log_10_SACort of NOV, but not of PRT (Table 3). Specifically, in response to NOV, log_10_SACort was higher in older animals compared to younger animals and in females compared to males.

Figure 3 depicts SACort in response to PRT depending on the animal handling method. In addition, the females, SA, AR and SF, were the dominant animals. The results of the respective LMM (Table 4) indicate that the effect of the handling method and female social status were not significant predictors of log_10_SACort in females.

Moreover, the LMMs indicated that the zoo was also not a significant predictor of the average log_10_SACort (Table 3 and Table 4).

Reproductive status had no significant effect on the log_10_SACort of females at the individual level (SA: LMM coefficient = −0.04, *SE* = 0.03, *t* = −1.62, *p* > 0.05; SW: LMM coefficient = 0.03, *SE* = 0.04, *t* = 0.66, *p* > 0.05, Figure 4).

On the individual level, the ovarian cycle phase had no significant effect on the log_10_SACort in SW (LMM coefficient = −7.1 × 10^−4^, *SE* = 0.03, *t* = −0.02, *p* > 0.05), CH (LMM coefficient = −0.04, *SE* = 0.02, *t* = −1.44, *p* > 0.05) and SF (LMM coefficient = 0.03, *SE* = 0.04, *t* = 0.77, *p* > 0.05). The effect was significant in SA (LMM coefficient = 0.07, *SE* = 0.03, *t* = 2.39, *p* < 0.05) and approached significance in TI (LMM coefficient = −0.07, *SE* = 0.03, *t* = −1.96, *p* = 0.056). SA had higher SACort levels in the luteal phase than during ovulation and the follicular phase. TI had lower levels in the luteal than in the follicular phase (Figure 5).

## 4. Discussion

Positive reinforcement training (PRT) and environmental enrichment are regularly used as “stress management tools” in captive elephant management. For example, by habituating animals to health checks through PRT or encouraging natural behaviors and avoiding boredom through enrichment, negative long-term stress can be effectively reduced and even prevented from occurring in the first place. The present study aimed to find out whether PRT and novel object exposure (NOV), as a form of environmental enrichment, induce a physiological response measured by salivary cortisol concentrations (SACort).

The results confirmed both of the following predictions (see Section 1.4): (i) that the SACort responses differed between PRT and NOV and (ii) that variation is related to specific individual parameters. First, the SACort responses indeed differed between PRT and NOV. This difference was related to both the average SACort level and the direction of SACort change over time. While PRT was followed by a small, but significant decline in the average SACort level, NOV induced a moderate but significant cortisol increase. As a consequence, SACort levels in response to PRT were on average lower than those in response to NOV. Second, the use of LMMs not only allowed us to identify these average trends by controlling intra- and inter-individual variation. The LMMs also provided information on the effect of factors determining individual variation in SACort. In this regard, we showed that age class and sex, but not zoo, handling method, social status or reproductive status in study females had an effect on SACort.

### 4.1. Effect of the Treatment on SACort

The SACort level in response to PRT was, on average, lower than that in relation to NOV. Since the SACort0 concentrations determined in samples collected immediately prior to PRT or NOV were similar, the difference in the mean SACort levels in response to the two treatments were likely due to the fact that PRT induced a decrease in the SACort, while NOV elicited an increase. The different reactions of the elephants towards the two situations can be explained by results from the literature. With regard to PRT, previous studies in primates [94,95] and African elephants [31] showed that routine PRT did not elicit a GC change and thus was not perceived as a stressor by the animals under study. Our findings of a small but significant decrease in SACort levels in response to PRT even suggest a potential stress-lowering effect of PRT in the African elephants studied. Such an effect is reasonable, as PRT, in general, allows the animal to control access to rewards through its behavior under voluntary cooperation and free choice of actions without fearing negative consequences [92]. Indeed, this perception of control is well-known to exert a stress-buffering and cortisol-lowering effect [53,110,111]. In addition, previous experience with PRT and thus its high predictability for the elephant may have caused an anticipatory response in the animals studied [53], i.e., relatively high pre-PRT (SACort0) levels compared to post-PRT SACort. PRT was part of the routine management of the elephants studied here and was indicated by the individual separation as well as the presence of the training tools and food buckets next to the training location. Therefore, based on the previous experience with PRT, the animals were probably able to predict the rewarding stimuli in the course of PRT, which resulted in anticipatory responses. According to [53], a GC time course characterized by an anticipatory response indicates that the situation was not perceived as a stressor by the animal. One can even assume that this anticipatory response indicates the positive affective state of anticipation [8]. To support this assumption, the anticipatory behaviors of the elephants can be recorded in the future [112,113].

By contrast, the small, but significant increase in SACort concentrations in response to NOV in the present study suggests that NOV was perceived as a mild stressor by the animals. This is not surprising since, compared with PRT, the perception of predictability and control was reduced in the NOV situation, where a different novel object was presented in each trial. The finding of an SACort increase in response to NOV is in line with the few previous studies on the effect of the presentation of novel objects on blood GC levels in other species (cattle: [86]; birds: [88,89]). Moreover, in the safe and protected zoo environment, novel objects likely only elicit moderate cortisol responses because the potential benefits (e.g., food acquisition) outweigh the potential costs (e.g., a hidden predator) of exploring novel objects. This applies especially to African elephants, a species that lacks natural predators [114]. Finally, in concert with the individual perception of predictability and control in NOV, certain affective states may have individually elicited SACort increases in certain animals [115]. However, considering SACort changes alone does not allow us to determine whether the affective states during NOV were more positive (e.g., arousal) or negative (e.g., anxiety) [8,115]. Similar to that in PRT, recording behavioral responses may shed light on this as well.

### 4.2. Effect of Individual Parameters on SACort in Relation to PRT and NOV

The present study found individual variation in SACort. This result is in line with the general finding across studies and species that some animals mount a cortisol response to an acute stressor or any situation that is not actually considered as a stressor (such as PRT in the present study). Other animals, in contrast, show a stress response with a lower increase in cortisol or a different time course, for example, a different peak time [42,43], or do not respond at all [41].

Since inter-individual differences in the GC levels between African elephants were also reported in previous studies [29,31,116,117], our results give further insights into the factors potentially explaining this inter-individual variation. First, age significantly contributed to variation in SACort, with younger animals having a lower SACort than older animals. This finding is in line with the results of previous studies of GC levels in blood or their metabolites in urine, respectively, in African elephants [22,57] and Asian female elephants [56] but contradicts the findings of blood GC levels in chimpanzees [42]. Second, we found a sex effect with male African elephants having a lower SACort than females. Evidence from studies of GC levels determined in blood, saliva or excreta in the same and other species partly supports (dwarf hamster: [43]; human: [118]) and partly contradicts the sex difference found in the present study (African elephant: [22]; great apes: [42,94]; spotted hyena: [47]; domestic pig: [119]). It has been proposed that the effect of sex steroids on HPA axis activity determines sex differences in GC responses [1]. To our knowledge, this relationship has not yet been shown in elephants, but is conceivable considering that there is an interaction between the reproductive status and HPA axis activity within sexes in both elephant species [57,61,62,63].

In the present study, reproductive status and ovarian cycle phase were checked as other individual factors influencing the SACort in females, because females were routinely monitored by urinary hormone analysis [99]. The reproductive status and/or ovarian cycle phase varied within animals over time. Based on the small and unbalanced sample size relative to the number of reproductive states or ovarian cycle phases that occurred in the present study, there was no significant variation in SACort associated with changes in reproductive status, contrary to the expectation based on findings in the literature (Asian elephant: [56,65]). The effect of the ovarian cycle phase that we detected in two of the five cows with an ovarian cycle was inconsistent, with only one cow tending (i.e., the effect only approached significance) to have lower cortisol levels in the luteal phase than in the follicular phase, as expected based on the literature (African: [62], Asian: [56,64]). Although the results of the present study are mixed, future studies using SACort as a stress response parameter should include variation in reproductive status and ovarian cycle phase and their interaction with sex and age [56] to account for potential effects on the SACort dynamics in captive African elephants.

Zoo and handling method did not contribute to the variation in SACort in the present study. Previously, an effect of zoo (i.e., sum of conditions describing the facility in which animals were housed, such as physical structure, enrichment opportunities and availability of social interaction) on serum GC levels was found in females of both elephant species [23]. The same study, however, could not demonstrate an effect of the handling method on serum cortisol levels in the animals. Moreover, SACort was not related to social status in the present study. This finding is consistent with a previous study in captive African elephants [120] in which no correlation of serum GC levels with social status was found.

It is conceivable that inter-individual differences in GC levels cannot be attributed to a single factor because there are many factors that can modulate individual GC levels and that can vary from animal to animal, including personality, life experience, genetic predisposition for low stress responsiveness, stressor type, social interaction and other environmental factors [1]. Future studies, focusing on the effect of the handling method on GC levels, for example, should monitor the same animals in both handling systems, that is when handling is changed from free to protected contact (the respective effect on behavior has been described by [121]). This would limit the number of factors differing between animals and a confounding of the potential effect of the type of elephant handling on cortisol levels.

### 4.3. Stress Management through Positive Reinforcement Training and Enrichment

The results of the present study show that positive reinforcement training and exposure to a novel object (as a surrogate for environmental enrichment) were, on average, not perceived as stressors or only as mild stressors, respectively, by the African elephants studied. The reaction towards PRT was probably characterized by a high level of predictability and controllability, which are psychological factors shown to lower stress [110,122] and assumed to underlie the animal welfare benefit of PRT [123]. NOV in the present study induced a short-term and small SACort increase of 10% in 2016 and 20% in 2017. Previous studies reported higher relative cortisol increases in response to different stressors: Dwarf hamsters responded with a relative cortisol increase of 45% to on-back restraint [43] and the average cortisol levels in cattle rose by 118% after social separation [36]. In [31], cortisol increased by 271%, 94% or 1833% in three African elephant cows in response to a mild social stressor. Therefore, we can assume that the low physiological response observed in the present study indicates mild stress in the elephants studied. Finally, in both PRT and NOV, the influence of affective states should be considered in interpreting the observed responses. While anticipation may have contributed to the response to PRT, the moderate SACort increase following NOV may have been determined by arousal (positive) and/or fear (negative) [8,115].

In addition, other stress responses besides increased HPA axis activity, such as behavioral adaptations and the reactions of the SNS, may have contributed to stress coping. The results presented in this paper do not allow us to draw conclusions about the interplay between behavioral and physiological stress responses. Still, we assume behavioral changes contributed to coping with potential stress, as we observed diverse behavioral responses to the novel object. The situation of NOV allows animals many behavioral options, which have been shown to promote stress management [70]. Moreover, the SNS may have supported stress coping as a function of the perception of the controllability of the situation [124].

In summary, the results of the present study allow conclusions about the effects of PRT and NOV on the individual stress coping abilities of captive African elephants. However, since these are only limited thus far, more comparative studies are needed to demonstrate how elephants in zoos with different levels of PRT and enrichment cope with potential stressors in captivity. These also include situations such as transfers, births, introduction of new elephants, etc.

### 4.4. Methodological Considerations and Limitations

The present study included elephants from three different zoos. This was associated with many potential confounding variables (e.g., enclosure, group structure, work routine, etc.) due to the different housing conditions and management systems. Whilst it was impossible to measure all these factors, they may still have influenced differences in cortisol levels and behavior between elephants and time points. However, a multi-zoo study per se comprises even more different study settings and is only possible with the given small number of animals per zoo. To minimize the influence of confounding variables on cortisol secretion, this work followed the guideline of altering the keepers’ work routines as little as possible to conduct the measurements under conditions familiar to the animals [125]. A good experimental design can minimize the influence of confounding variables [125]. For example, only adult elephants were studied. In addition, the constellations of the groups in Kronberg and Erfurt were similar (one bull, two cows). Finally, each animal served as its own control. The cortisol levels before and after a PRT and NOV trial were always paired with individuals and trials. Such a repeated measures design can avoid confounding variables related to differences between animals that may influence the outcome of an experiment [126,127].

The approach described above of integrating the study into zoo-specific daily routines resulted in differences between zoos in the time of day at which the experiments and saliva collections were performed. This was not optimal given the diurnal variation in cortisol secretion [54], which also occurred in our study animals [30], and its interaction with stress-induced cortisol secretion [128]. For example, for the trials conducted between 13:00 and 14:00, cortisol is expected to decrease slightly as part of the diurnal rhythm [58,60] and this may have potentially confounded our findings of salivary response pattern to our experimental trials. This was less expected for trials conducted at 18:00 when diurnal cortisol levels are expected to be at their nadir [58,60]. By including the factor “zoo” in the LMMs, however, we controlled for the difference in sampling times between zoos with zoo having no effect on SACort levels. Within the zoos and elephants, the sampling times were constant. Since the diurnal decline in cortisol concentrations during the time interval of our experimental testing was expected to be rather small [58,60] and, moreover, the average cortisol response to our treatments was, at least for NOV, in the opposite direction (Figure 1), the sampling time was unlike to affect our findings on cortisol responses to the experimental treatments, at least not to a great deal.

Saliva was collected every 10 or 15 min after PRT or NOV, respectively, depending on the zoo. This temporal resolution is consistent with the interval chosen by previous studies to examine salivary cortisol responses to stressors [36,37,38,129]. After the Kronberg study, where the 10-min sampling interval was implemented, we decided to use the 15-min sampling interval in the remaining two zoos to reduce the sampling effort. We considered this longer sampling interval sufficient for the purposes of this study. Therefore, we recommend that pilot studies be conducted in future studies to assess, among other things, the feasibility of the sampling schedule within the work routines of the participating zoos.

We decided for a maximum of a 60-min sampling period because a stressor-induced cortisol increase in saliva occurs within this period in the majority of species (human: [35]; cattle: [36]; primates: [37,130]; dog: [38]; horse: [40]), although longer lag times are sometimes reported, e.g., [39]. For animal welfare reasons, we decided not to prolong the time of separation of the animals just for the reason of an extended time of sample collection. Indeed, the individual and average SACort response patterns to PRT and NOV of our study animals suggest that elephants do also fall within the 60-min range of salivary cortisol lag times. We, therefore, believe that the 1-h sampling period applied here was appropriate for testing the predictions and achieving the aims of the present study. If studies require detailed information on the exact time course of the SACort responses in African elephants, conducting an ACTH challenge test, e.g., [37], in combination with an extended period of repeated sampling would be useful. Such knowledge may also benefit future studies that focus on examining the variation in the time courses and duration of GC responses, as both the magnitude and the duration of the GC release determine if the stress response affects welfare [14].

The number of trials to test the effect of PRT on SACort was relatively small compared to NOV. We cannot exclude that a higher number of repetitions of PRT could have led to a stronger effect. However, the mean effect of NOV on SACort was also relatively small but still significant, as was the effect of PRT on SACort. Therefore, we are confident that the total of 30 PRT trials in ten animals was sufficient to obtain meaningful data.

The PRT and each saliva sampling event included rewarding the elephants with small pieces of bread, which ensured the animals’ cooperation. Additionally, during the post-sampling period, they were offered small amounts of hay or hay pellets to keep the animals occupied, with the intention of preventing a rise in cortisol due to the relatively long stay in the box separated from the herd. We cannot exclude that food consumption had an effect on salivary cortisol levels [100,101,102]. However, this may not necessarily be the case [131,132]. Nevertheless, we recommend regulating food and water intake during the sampling period to ensure consistency in this experimental condition.

The present work focuses on the difference in the change in salivary cortisol levels between PRT and NOV. Behavioral observations would have been a useful addition to determine how elephants perceive PRT or NOV. Future studies addressing this question should, for example, analyze anticipatory behavior and behavioral responses to the novel object, such as fear displays or play, to test whether behavior reflects cortisol changes.

## 5. Conclusions

While positive reinforcement training (PRT) probably acts as a stress-buffering event lowering salivary cortisol (SACort), exposure to a novel object (NOV) (as a form of environmental enrichment) may have been perceived as a mild stressor. These “stress management tools” thus have different effects on the physiological stress levels of captive African elephants and the impacts are more likely to be positive, in the sense that they may evoke positive affective states, such as anticipation and arousal, as well as support the animals’ welfare and stress coping abilities. Definite conclusions in this regard, however, are challenging based on the results presented here. Complementary behavioral studies could shed light on the individual perception of PRT and NOV and, therefore, potential individual welfare effects. In addition, further studies comparing elephants kept under different PRT and enrichment regimes could allow firm conclusions regarding the effect of PRT and NOV on stress coping abilities.

Age and sex, but not zoo, handling method, reproductive status or social status, determined inter-individual differences in SACort in the captive African elephants studied. Thus, the present study again demonstrated the individuality of the GC response. It also illustrated the complexity of the potential factors, including affective states, which influence the perception of and the physiological response to potentially stressful situations in the African elephant. Understanding the individuality of stress in the African elephant is worth striving for in order to support management decisions aimed at promoting the individual welfare of this highly demanding zoo animal species.

## Figures and Tables

**Figure 1 animals-11-03525-f001:**
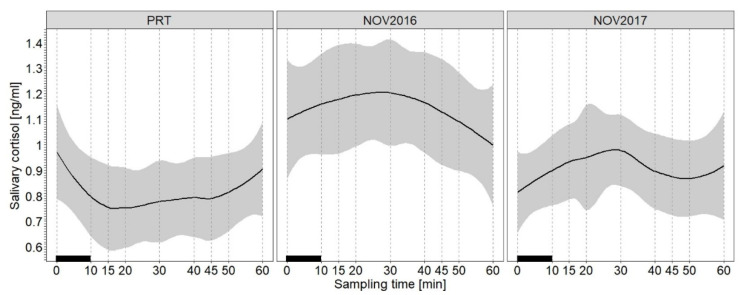
Average salivary cortisol time courses across sampling times (0–60 min) including 95% confidence intervals (shaded areas) in response to positive reinforcement training (PRT) and novel object exposure in the first (NOV 2016) and second study period (NOV 2017). Black bars indicate the time period in which PRT and NOV, respectively, occurred. LOESS smoothing (solid line) was applied to salivary cortisol concentrations conditional on time.

**Figure 2 animals-11-03525-f002:**
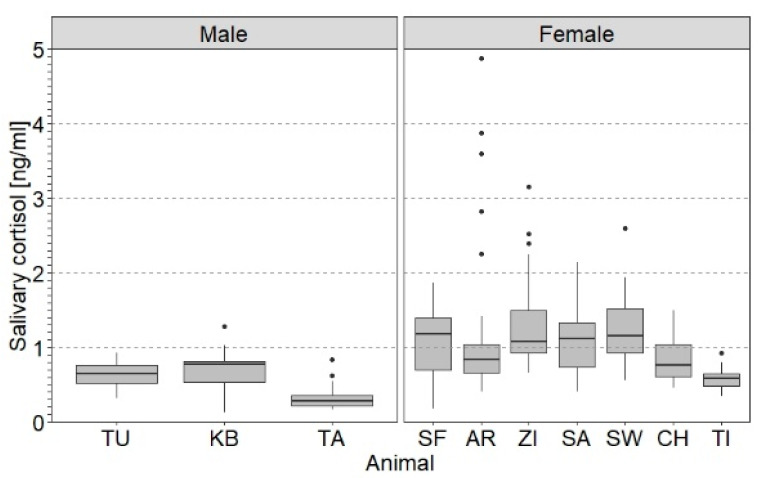
Box plots of salivary cortisol levels (SACort) in response to novel object exposure (NOV) of male (TU, KB and TA) and female (SF, AR, ZI, SA, SW, CH and TI) captive African elephants in the second study period (2017). The boxes illustrate the 25% and 75% quartiles, bars indicate medians, dots indicate outliers. Sample sizes per animal include total number of samples collected in NOV 2017 in the respective animal and are given in Appendix A. Within sexes, animals are sorted by decreasing age from left to right.

**Figure 3 animals-11-03525-f003:**
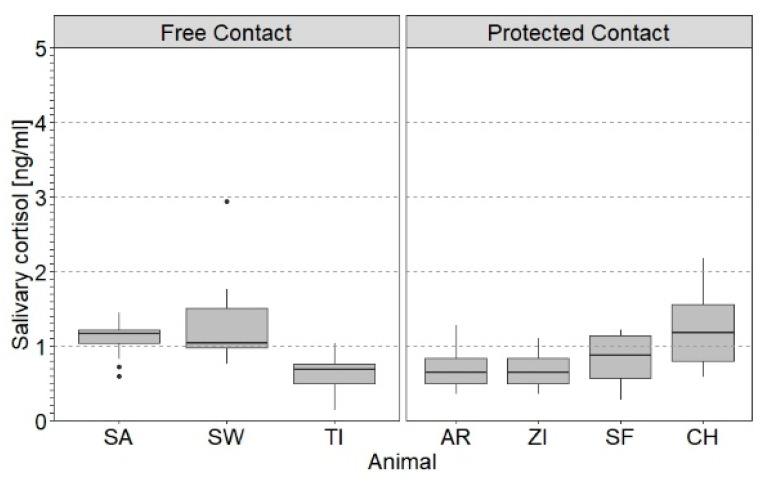
Box plots of salivary cortisol levels (SACort) in relation to handling method and zoo of female African elephants. Free contact in females kept in Wuppertal (SA, SW and TI), protected contact in females housed in Kronberg (AR and ZI) and Erfurt (SF and CH). Shown are SACort in relation to positive reinforcement training (PRT). The boxes illustrate the 25% and 75% quartiles, bars indicate medians, dots indicate outliers. Sample sizes per animal include total number of samples collected in PRT in the respective animal and are given in Appendix A.

**Figure 4 animals-11-03525-f004:**
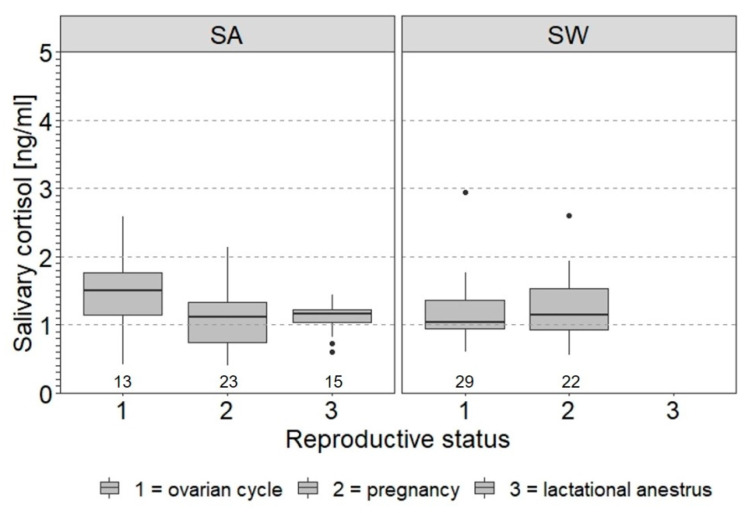
Box plots of salivary cortisol levels (SACort) in relation to reproductive status of female African elephants SA and SW. Facets each show individual data of females that exhibited ovarian cycles at some point during the study. The boxes illustrate the 25% and 75% quartiles, bars indicate medians, dots indicate outliers. Sample size per animal and reproductive status is given below the boxes. Data used include SACort values of both treatments and study periods. See Table 1 for information on reproductive status of females.

**Figure 5 animals-11-03525-f005:**
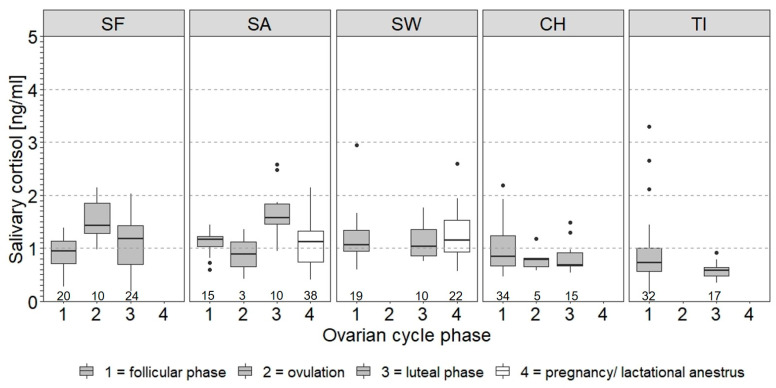
Box plots of salivary cortisol levels (SACort) in relation to ovarian cycle phase of female African elephants. Facets each show individual data of females that exhibited ovarian cycles at some point during the study. The boxes illustrate the 25% and 75% quartiles, bars indicate medians, dots indicate outliers. Sample size per animal and ovarian cycle phase is given below the boxes. Animals are sorted by decreasing age from left to right. Data used include SACort values of both treatments and study periods. See Table 1 for information on ovarian cycle phase of females.

**Table 1 animals-11-03525-t001:** Information on the animals included in the present study.

Zoo	Animal-ID	Sex ^†^	Age Class ^‡^	Handling Method ^§^	Social Status ^¶^	Reproductive Status of Females ^#^	Ovarian Cycle Phase ^#^
Kronberg	AR	F	3	PC	High	2016, 2017: acyclic	--
	ZI	F	3	PC	Low	2016, 2017: acyclic	--
	TA	M	1	PC	Lowest *	--	--
Wuppertal	SA	F	2	FC	High	2016: lactational anestrus, cyclic 2017: pregnant	2016: ovulation, luteal 2017: --
	SW	F	2	FC	Low	2016: cyclic 2017: pregnant	2016: luteal, follicular 2017: --
	TI	F	1	FC	Low	2016, 2017: cyclic	2016: follicular 2017: luteal, follicular
	TU	M	2	PC	Highest *	--	--
Erfurt	SF	F	3	PC	High	2016, 2017: cyclic	2016: follicular, ovulation, luteal 2017: ovulation, luteal
	CH	F	1	PC	Low	2016, 2017: cyclic	2016: follicular, ovulation, luteal 2017: follicular
	KB	M	1	PC	In between *	--	--

^†^ F = female, M = male, ^‡^ 1 = 5–19.9 years (adolescent/young adult), 2 = 20–34.9 years (adult), 3 = 35–49.9 years (older adult) [98], ^§^ PC = protected contact: keepers always interact with the elephants through a barrier, FC = free contact: keepers enter the elephant enclosure and handle the elephants freely without a protecting barrier, ^¶^ The animal is dominant or not, judged based on displacements from certain positions, e.g., feeding sites (judged by keepers), High = dominant female in the respective group, Low = subordinate female in the respective group, Highest/Lowest/In between * = Male has highest or lowest rank or ranks between dominant and subordinate female when temporarily with the females, ^#^ Females in all zoos were routinely monitored non-invasively for reproductive status by urinary hormones analyzed in weekly samples as described in [99]. Categories according to [56], ovulation was added as an additional category. Information on reproductive status of males (e.g., musth phase) was not available.

**Table 2 animals-11-03525-t002:** Overview of experimental and individual parameters fitted as fixed effects in the linear mixed models (LMM). PRT = positive reinforcement training, NOV = novel object exposure.

Fixed Effects		LMM Name				
		All Animals/PRT and NOV	All Animals/PRT	All Animals/NOV	Females/PRT	Females/NOV
Experimental	Situation	**Yes**	No	No	No	No
	Study period ^†^	**Yes**	No	**Yes**	No	**Yes**
	Sampling time	**Yes**	**Yes**	**Yes**	**Yes**	**Yes**
Individual	Age class	**Yes**	**Yes**	**Yes**	**Yes**	**Yes**
	Sex	**Yes**	**Yes**	**Yes**	No	No
	Zoo	**Yes**	**Yes**	**Yes**	**Yes**	**Yes**
	Social status ^‡^	No	No	No	**Yes**	**Yes**
	Handling method ^‡^	No	No	No	**Yes**	**Yes**

^†^ Only in NOV **^, ‡^** Only in females.

**Table 3 animals-11-03525-t003:** Fixed effects resulting from the linear mixed model examining the effect of experimental (sampling time [linear term]: 0, 10, 15, 20, 30, 40, 45, 50, 60 min, sampling time sq [quadratic term: ${\left(\mathrm{sampling}\;\mathrm{time}-30\right)}^{2}$], study period (only novel object exposure [NOV]) and individual (age class, sex, zoo, see Table 1) factors on the salivary cortisol levels of ten captive African elephants in relation to positive reinforcement training (all animals/PRT, *n* = 167) or NOV (all animals/NOV, *n* = 422). Salivary cortisol concentrations were log_10_-transformed to satisfy the model assumptions (see Section 2.5). Animal-ID (both models: *n* = 10) and trial nested in animal-ID (PRT: *n* = 30, NOV: *n* = 60) were included as random effects.

**All Animals/PRT**					
**Fixed Effect**	**Estimate**	** *SE* **	**df**	** *t* **	** *p* **
**Intercept**	**−0.76**	**0.28**	**6.00**	**−2.72**	**0.035**
Sampling time	−5.36 × 10^−4^	5.28 × 10^−4^	135.30	−1.02	0.312
**Sampling time sq**	**7.33 × 10^−5^**	**3.00 × 10^−5^**	**135.30**	**2.44**	**0.016**
Age class	0.06	0.08	6.00	0.81	0.449
Sex	0.12	0.14	6.02	0.86	0.422
Zoo	0.14	0.08	5.96	1.79	0.125
**All Animals/NOV**					
**Fixed effect**	**Estimate**	** *SE* **	**df**	** *t* **	** *p* **
**Intercept**	**−0.53**	**0.13**	**6.77**	**−4.17**	**0.005**
Sampling time	1.23 × 10^−4^	4.19 × 10^−4^	373.70	0.29	0.39
**Sampling time sq**	**−3.76 × 10^−5^**	**2.23 × 10^−5^**	**360.40**	**−1.69**	**0.05**
**Study period**	**−0.11**	**0.02**	**388.00**	**−6.34**	**<0.001**
**Age class**	**0.11**	**0.04**	**6.11**	**3.10**	**0.02**
**Sex**	**0.20**	**0.06**	**6.18**	**3.27**	**0.02**
Zoo	0.05	0.03	5.88	1.37	0.22

Note: Two-sided *t*-test (Satterthwaite’s method) were used to test significance of fixed effects except for sampling time and sampling time sq in NOV. In these fixed effects, one-sided *t*-tests (Satterthwaite’s method) were used. Statistical significance (*p* < 0.05) in bold.

**Table 4 animals-11-03525-t004:** Fixed effects resulting from the linear mixed model examining the effect of experimental (sampling time [linear term]: 0, 10, 15, 20, 30, 40, 45, 50, 60 min, sampling time sq [quadratic term: ${\left(\mathrm{sampling}\;\mathrm{time}-30\right)}^{2}$], study period (only novel object exposure [NOV])) and individual (age class, handling method, social status and zoo, see Table 1) factors on the salivary cortisol level of seven female captive African elephants (PRT: *n* = 116, NOV: *n* = 293). Salivary cortisol concentrations were log_10_-transformed to satisfy the model assumptions (see Section 2.5). Animal-ID (both models: *n* = 7) and trial nested in animal-ID (PRT: *n* = 21, NOV: *n* = 42) were included as random effects.

**Females/PRT**					
**Fixed Effect**	**Estimate**	** *SE* **	**df**	** *t* **	** *p* **
Intercept	−0.27	0.64	1.99	−0.42	0.713
Sampling time	−7.72 × 10^−4^	6.29 × 10^−4^	93.13	−1.23	0.223
**Sampling time sq**	**1.20 × 10^−4^**	**3.57 × 10^−5^**	**93.11**	**3.37**	**0.001**
Age class	0.03	0.15	1.99	0.22	0.848
Handling	−0.09	0.18	1.99	−0.48	0.681
Social status	0.02	0.19	1.97	0.12	0.918
Zoo	0.09	0.12	1.98	0.78	0.518
**Females/NOV**					
**Fixed effect**	**Estimate**	** *SE* **	**df**	** *t* **	** *p* **
Intercept	−0.06	0.24	1.99	−0.27	0.814
**Sampling time**	**9.37 × 10^−4^**	**5.34 × 10^−4^**	**261.10**	**1.75**	**0.040**
**Sampling time sq**	**−6.27 × 10^−5^**	**2.85 × 10^−5^**	**250.20**	**−2.20**	**0.015**
**Study period**	**−0.12**	**0.02**	**273.90**	**−5.17**	**<0.001**
Age class	0.11	0.06	1.88	1.99	0.194
Handling	−0.04	0.07	1.89	−0.57	0.629
Social status	0.04	0.07	1.82	0.64	0.594
Zoo	0.01	0.04	1.82	0.24	0.838

Note: Two-sided *t*-test (Satterthwaite’s method) were used to test significance of fixed effects except for sampling time and sampling time sq in NOV. In these fixed effects, one-sided *t*-tests (Satterthwaite’s method) were used. Statistical significance (*p* < 0.05) in bold.

## Data Availability

We hereby confirm that the data set supporting the conclusions of this article will be made available in a publicly accessible repository (Dryad) after publication.

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
