# Peer review of "Effects of Positive Reinforcement Training and Novel Object Exposure on Salivary Cortisol Levels under Consideration of Individual Variation in Captive African Elephants (Loxodonta africana)"

_animals, 2021, doi:10.3390/ani11123525_

Round 1

Reviewer 1 Report

Dear authors,

I thought this was an interesting paper, nicely written and easy to read. My main concern (which is a theme throughout my comments – sorry to keep repeating it but I just wanted to make sure I had flagged the relevant places) is the use of the word ‘stress’ and the lack of consideration that physiological parameters can also indicate a positive arousal. Indeed, this point is probably supported by your own findings when you had increased cortisol before the training sessions – likely anticipation of a positive event. the use of the word ‘stress’ implies a negative valence and so I think this needs reconsidering throughout the MS. It is important that we consider physiological parameters alongside other measures in order to be able to say whether this was positive or negative, especially when we are dealing with such small increases. Did you gather data on behavioural responses at all? if so this would be really useful in terms of indicating possible anticipatory behaviour or looking at individual responses to the novel object. I think it would also be beneficial to just add a note into the methods which says that the procedures were ended if the elephants were considered to be overtly stressed (which I assume they weren’t), it’s just useful to highlight, especially when they were being separated for an hr.

Line by line comments below:

L19: I don’t think stress is the correct word here – you were looking at physiological parameters but stress implies a negative valence

L21: for the reasons above, I don’t think ‘stress hormone’ is needed – simply sampling of salivary cortisol is sufficient, or you could say which can be used as a measure of stress or stimulation/arousal. I know this a simple summary but I think including this suggests that increased cortisol is always a negative thing, which is incorrect. 

L26: how are you defining ‘little’? why is there no consideration the the increased salivary cortisol after enrichment may be indicative of arousal?

Abstract: it would be useful to say in here how many samples you actually had – ‘repeatedly sampled’ is a bit vague and not particularly helpful to the reader

L63: latin name needed for African elephants

L157: I am not clear why you are linking enrichment with the concept of short-term adaptive stress, and why there is no consideration of enrichment being a stimulant which engages animals

Table 1: how did you define social status?

L213: occasional

L248: so they had two of the novel objects presented at the time or did they have two versions of the novel object presented? Please clarify this, and explain why this decision was made.

L263: please specify how many of these samples were pre and how many samples were post PRT

L451: none of your values in Table 4 are in italics so the last part of that sentence can be removed

Table 3: I am not convinced this entire table is needed – the main result that seems to be being presented in this is the results in relation to situation (as everything else is presented in T4), I think if it is just this line then this table would be better being deleted and that information including in the text. As the results were quite different for PRT and NOV it doesn’t seem to make sense to have a combined model for these situations anyway, so T4 is quite clear in terms of factors affecting salivary cort in the two quite separate situations.

L502/503: there are many other reasons beyond ‘stress management’, and I think they should be included here. PRT is used to help make sure animals can have health checks without the need for invasive procedures and enrichment is used to stimulate animals/prevent boredom, to make their enclosures more naturalistic etc.

L541 onwards: Again here it isn’t all about stress as a negative point, it can also show excitement/arousal, so the novel object may have been ‘exciting’ for the elephants rather than a negative effect. Did you gather any behavioural data during these observations? Behavioural data would potentially support your anticipatory theory (which I would think is the case as elephants often anticipate scheduled events) and it would also allow you to look at latency to approach novel objects, time spent with the novel object etc. it’s important that cortisol isn’t used alone as a measure of ‘stress’ and I think this is something that as a field we should be moving away from – because of not being able to say whether the valence of the event is positive or negative. Combining behavioural data would support this work significantly.

L620: indicating a physiological response to the situation – possibly arousal, not just stress as a negative situation.

L628 onwards: I don’t understand how you have jumped to this conclusion, nor how your work supports this. you were specifically looking at physiological responses to two situations but that doesn’t relate to ‘stress coping options’, and you have not reported any information on how the animals were behaving or even qualitative assessment of their demeanour to support this.

L687: see my comments above – please recognise the potential arousal aspects

L690: as above, I don’t think you have evidence for this point

L694: again, this needs to be considered in terms of stress vs arousal – you haven’t looked at perception of stress here, you have looked at physiological responses to situations. Which MAY be a stressor, but equaly may not. I think this sentence needs reconsidering.  

Author Response

Dear Reviewer 1,

Thank you very much for your very helpful comments. We can understand all the comments and have incorporated them into the revised manuscript. We now consider arousal and other affective states in the context of our results and have also refined the definition of stress. The answers to your comments (in bold) can be found directly below the respective comments. The line numbers refer to the "all markups displayed" option.

Dear authors,

I thought this was an interesting paper, nicely written and easy to read. My main concern (which is a theme throughout my comments – sorry to keep repeating it but I just wanted to make sure I had flagged the relevant places) is the use of the word ‘stress’ and the lack of consideration that physiological parameters can also indicate a positive arousal. Indeed, this point is probably supported by your own findings when you had increased cortisol before the training sessions – likely anticipation of a positive event. the use of the word ‘stress’ implies a negative valence and so I think this needs reconsidering throughout the MS. It is important that we consider physiological parameters alongside other measures in order to be able to say whether this was positive or negative, especially when we are dealing with such small increases. Did you gather data on behavioural responses at all? if so this would be really useful in terms of indicating possible anticipatory behaviour or looking at individual responses to the novel object.

I think it would also be beneficial to just add a note into the methods which says that the procedures were ended if the elephants were considered to be overtly stressed (which I assume they weren’t), it’s just useful to highlight, especially when they were being separated for an hr.

L373-374: As recommended, we added a note in the methods that the procedures and sampling were stopped if the animals were considered to be overtly stressed.

L901-907: We have also added to the Institutional Review Board Statement at the end of the manuscript in this regard.

And yes, the elephants did not appear to be overtly stressed, highly likely because the training and saliva sampling was positively reinforced. In the case of NOV, they had the possibility to retreat into the box away from the object.

Reply to your comments regarding the definition and use of “stress” and lack of consideration of arousal (or affective states in general). OKAY

We used the neutral definition of stress in this paper. Therefore, cortisol stress responses are neither positive nor negative in the first instance. They just show that the organism copes with the state of stress.

L55-67: We added a more detailed definition of stress and explanation of the valence of stress.

L315-319: We highlight the non-valence of the stress in the present study.

We now consider that relatively high GC levels can also indicate the positive affective states of arousal and/or anticipation.

L27-30 and L46-48: We mention in the abstract that the novel object may have elicited arousal (and PRT may have associated with anticipation

L77-79: We included the influence of affective states (e.g. arousal) on GC secretion.

L650-653 and L755-757: We consider that the relatively high cortisol before PRT may indicate the positive affective state of anticipation and that behavioral data may shed light on this (see also below your comment referring to line 541).

L664-669 and L755-757: We consider that the cortisol increase after NOV may indicate arousal, also here behavioral day may shed light on this (see also below your comment referring to line 541).

Finally, as mentioned above, we also address the fact that behavioral data, together with cortisol levels, can provide information about affective state and explain why we did not includ behavioral data in the present manuscript.

L758-769: We point out that behavioral measures may have been contributed to stress coping.

L852-856: We point out behavioral measures are useful to judge how elephants perceived the treatments.

We refer to this general reply in your line by line comments regarding stress and arousal.

Line by line comments below:

L19: I don’t think stress is the correct word here – you were looking at physiological parameters but stress implies a negative valence

L19-20: We have used a more neutral term here and say now “physiological changes”.

L21: for the reasons above, I don’t think ‘stress hormone’ is needed – simply sampling of salivary cortisol is sufficient, or you could say which can be used as a measure of stress or stimulation/arousal. I know this a simple summary but I think including this suggests that increased cortisol is always a negative thing, which is incorrect. 

L21-22:We changed the wording as requested.

L26: how are you defining ‘little’? why is there no consideration the the increased salivary cortisol after enrichment may be indicative of arousal?

L744-757: “Little” means: compared to cortisol responses reported by other studies. However, we changed the respective sentence in the abstract and “little” is removed.

L28-30: Now we mention that that the novel object may have elicited arousal (and PRT may have associated with anticipation).

Abstract: it would be useful to say in here how many samples you actually had – ‘repeatedly sampled’ is a bit vague and not particularly helpful to the reader

L40: We added the number of repeats in brackets behind “PRT” and “NOV”.

L63: latin name needed for African elephants

L88: We added the Latin name for African elephants.

L157: I am not clear why you are linking enrichment with the concept of short-term adaptive stress, and why there is no consideration of enrichment being a stimulant which engages animals

L175-195: We made some changes in this paragraph in order to clarify the link between short-term adaptive stress ad enrichment and to consider the stimulating effect of enrichment.

L181-182: Consideration of the stimulating effect of enrichment.

Table 1: how did you define social status?

L243-244: We added the definition: The animal is dominant or not, judged based on displacements from certain positions, e.g., feeding sites (judged by keepers).

L213: occasional

L257: Corrected.

L248: so they had two of the novel objects presented at the time or did they have two versions of the novel object presented? Please clarify this, and explain why this decision was made.

L296-299: We clarify and explain why two novel object were provided in the dyadic condition.

L263: please specify how many of these samples were pre and how many samples were post PRT

L375-383: We added this information a bit later in the text. You can also find the number of samples collected at each time point in the supplemental material.

L451: none of your values in Table 4 are in italics so the last part of that sentence can be removed

L525-526: We removed this part of the sentence.

Table 3: I am not convinced this entire table is needed – the main result that seems to be being presented in this is the results in relation to situation (as everything else is presented in T4), I think if it is just this line then this table would be better being deleted and that information including in the text. As the results were quite different for PRT and NOV it doesn’t seem to make sense to have a combined model for these situations anyway, so T4 is quite clear in terms of factors affecting salivary cort in the two quite separate situations.

L482: We agree. So we added just the values of the fixed effect Situation in the text. We moved Table 3 in the supplemental material (now Table S2).

L502/503: there are many other reasons beyond ‘stress management’, and I think they should be included here. PRT is used to help make sure animals can have health checks without the need for invasive procedures and enrichment is used to stimulate animals/prevent boredom, to make their enclosures more naturalistic etc.

L606-609: We now mention the advantages of PRT and enrichment that you stated.

L541 onwards: Again here it isn’t all about stress as a negative point, it can also show excitement/arousal, so the novel object may have been ‘exciting’ for the elephants rather than a negative effect. Did you gather any behavioural data during these observations? Behavioural data would potentially support your anticipatory theory (which I would think is the case as elephants often anticipate scheduled events) and it would also allow you to look at latency to approach novel objects, time spent with the novel object etc. it’s important that cortisol isn’t used alone as a measure of ‘stress’ and I think this is something that as a field we should be moving away from – because of not being able to say whether the valence of the event is positive or negative. Combining behavioural data would support this work significantly.

See above: general reply

We did not record anticipatory behavior. We did record behavioral responses to the novel object. Three elephants were very hesitant to approach novel objects while three other elephants always played with the novel object. The remaining three animals showed a more variable behavioral response. However, the relationship between the behavioral type to cortisol levels was not significant. Considering the considerable length of the manuscript and that we only recorded behavior in NOV, we decided to remove that part and to focus solely on the difference in the cortisol response between PRT and NOV.

L650-653 and L755-757: We consider that the relatively high cortisol before PRT may indicate the positive affective state of anticipation and that behavioral data may shed light on this.

L664-669 and L755-757: We consider that the cortisol increase after NOV may indicate arousal, also here behavioral day may shed light on this.

L758-769: We discuss the effects of other channels of the stress response: behavior and SNS.

L852-856: We added that behavioral observations would have been useful in the methodological considerations section.

L620: indicating a physiological response to the situation – possibly arousal, not just stress as a negative situation.

L744-752: We illustrated that low increases indicate mild stress by adding examples from other studies.

L752-757: We considered that the small increase may also indicate arousal.

L628 onwards: I don’t understand how you have jumped to this conclusion, nor how your work supports this. you were specifically looking at physiological responses to two situations but that doesn’t relate to ‘stress coping options’, and you have not reported any information on how the animals were behaving or even qualitative assessment of their demeanour to support this.

L770-775: We have reworded our conclusion to indicate that our results do not provide a firm statement regarding the effects of PRT and NOV on stress coping abilities.

L687: see my comments above – please recognise the potential arousal aspects

L866: We considered that PRT and NOV may evoke positive affective states, like arousal.

L690: as above, I don’t think you have evidence for this point

L863-872: We reworded this sentence and added that definite conclusions would require measurements of other parts of the stress response, like behavior.

L694: again, this needs to be considered in terms of stress vs arousal – you haven’t looked at perception of stress here, you have looked at physiological responses to situations. Which MAY be a stressor, but equaly may not. I think this sentence needs reconsidering.  

L875-878: We have rephrased this sentence to mention affective states such as arousal as factors influencing the physiological response (e.g., the cortisol response).

Reviewer 2 Report

This paper is well-written, reads smoothly and is clear. However it is not concise, but that is not of concern to the content. I could find no corrections to be made.

Author Response

Dear Reviewer 2,

Thank you very much for the positive feedback.

Reviewer 3 Report

The paper is well-written and includes a succinct review of stress physiology before presenting results from a study designed to test potential effects of positive reinforcement training and novel object exposure on salivary cortisol levels in a group of 10 African elephants (3 males and 7 females) housed in 3 European zoos. Because so many intrinsic and extrinsic factors can influence salivary concentrations of cortisol, it is important to minimize variables in study design. While the authors did pay attention to many variables, establishing baseline cortisol levels across one or more diurnal cycles for individual elephants was not done. This would have allowed for the determination of diurnal cortisol profiles and some measure of estrous cyclicity; both factors are known to significantly affect cortisol levels. Moreover, diurnal profiles may provide some measure of an individual’s health, overall stress level (i.e., robust or flat profile), and ability to respond to novel stress. Many limitations were appropriately acknowledged in the discussion, and this paper does contribute to our knowledge of cortisol physiology in captive African elephants, which is challenging to study.

Section 1.2 reviews factors affecting individual variation of glucocorticoid levels, including age, coping style, etc. but diurnal cycles are not mentioned! A note about this should be included, Some helpful references include:

  • Bechert et al. “Diurnal variation in serum concentrations of cortisol in captive African (Loxodonta africana) and Asian (Elephas maximus) elephants” (2021) Zoo Biol 40:458–471.
  • Plangsangmas et al. “Circadian rhythm of salivary immunoglobulin A and associations with cortisol as a stress biomarker in captive Asian elephants (Elephas maximus) (2020) Animals 10, 157; doi:10.3390/ani10010157.
  • Menargues et al. “Circadian rhythm of salivary cortisol in Asian elephants (Elephas maximus): a factor to consider during welfare assessment” (2012) J Appl Anim Welf Sci 15(4):383-90.

Were enough treatments applied to test for effects of positive reinforcement (3 trials per animal)?  Would additional trials have potentially resulted in more of an effect?  (There were 9 trials per animal for the novel object study.) Consider mentioning this in the discussion and interpretation of results.

Saliva samples were collected at 1300 and 1400 hours at two zoos when diurnal cortisol levels would have been declining, and at 1800 hours at one zoo when diurnal cortisol levels would be at their nadir. How this might have affected results should be described.

In the discussion paragraph reviewing effects of reproductive status on cortisol levels, it might be helpful to share that serum concentrations of progesterone and cortisol were positively correlated (p<0.01) in a small group of captive female African elephants (ref 98). Importantly, in addition to age, sex, reproductive status, handling method, and location, the discussion should include thoughts on how diurnal fluctuations in salivary cortisol levels may have impacted findings – especially given the different collection time in Kronberg.

While the paper is generally well-written, a few sentences would benefit from some grammatical revisions – e.g., lines 73-74: change “In zoo animals, by contrast, it can be trained as part of daily routines” to “Zoo animals, by contrast, can be trained for sample collections as part of daily routines”.

Author Response

Dear Reviewer 3,

Thank you very much for your very helpful comments. We can understand all the comments and have incorporated them into the revised manuscript. We now consider the diurnal variation and reproductive status as additional factors influencing cortisol secretion which was your main concern. The answers to your comments (in bold) can be found directly below the respective comments. The line numbers refer to the "all markups displayed" option.

The paper is well-written and includes a succinct review of stress physiology before presenting results from a study designed to test potential effects of positive reinforcement training and novel object exposure on salivary cortisol levels in a group of 10 African elephants (3 males and 7 females) housed in 3 European zoos. Because so many intrinsic and extrinsic factors can influence salivary concentrations of cortisol, it is important to minimize variables in study design. While the authors did pay attention to many variables, establishing baseline cortisol levels across one or more diurnal cycles for individual elephants was not done. This would have allowed for the determination of diurnal cortisol profiles and some measure of estrous cyclicity; both factors are known to significantly affect cortisol levels. Moreover, diurnal profiles may provide some measure of an individual’s health, overall stress level (i.e., robust or flat profile), and ability to respond to novel stress. Many limitations were appropriately acknowledged in the discussion, and this paper does contribute to our knowledge of cortisol physiology in captive African elephants, which is challenging to study.

Section 1.2 reviews factors affecting individual variation of glucocorticoid levels, including age, coping style, etc. but diurnal cycles are not mentioned! A note about this should be included, Some helpful references include:

  • Bechert et al. “Diurnal variation in serum concentrations of cortisol in captive African (Loxodonta africana) and Asian (Elephas maximus) elephants” (2021) Zoo Biol 40:458–471.
  • Plangsangmas et al. “Circadian rhythm of salivary immunoglobulin A and associations with cortisol as a stress biomarker in captive Asian elephants (Elephas maximus) (2020) Animals 10, 157; doi:10.3390/ani10010157.
  • Menargues et al. “Circadian rhythm of salivary cortisol in Asian elephants (Elephas maximus): a factor to consider during welfare assessment” (2012) J Appl Anim Welf Sci 15(4):383-90.

We reported gross diurnal changes in cortisol in our study animals in Hambrecht et al. (2020) Zoo Biol. Therefore, we now consider the effect of this diurnal variation on the results of the present study.

L143-145, 152 (section 1.2): We include day time as another factor that also contributes to individual variation in GC levels.

Were enough treatments applied to test for effects of positive reinforcement (3 trials per animal)?  Would additional trials have potentially resulted in more of an effect?  (There were 9 trials per animal for the novel object study.) Consider mentioning this in the discussion and interpretation of results.

L838-843 (section4.4): We agree that the number of tests for testing the effect of PRT on salivary cortisol are relatively small compared to the ones used for studying the effect of NOV on HPA-axis activity. We are nevertheless confident that the results of 30 trials carried out in total across the ten study animals provide meaningful data. However, as suggested by the reviewer we acknowledge this possible limitation of our approach in the discussion.

Saliva samples were collected at 1300 and 1400 hours at two zoos when diurnal cortisol levels would have been declining, and at 1800 hours at one zoo when diurnal cortisol levels would be at their nadir. How this might have affected results should be described.

L798-813: We discussed the potential effect of the diurnal variation on the findings and conclude that there may have been a small effect, if at all.

In the discussion paragraph reviewing effects of reproductive status on cortisol levels, it might be helpful to share that serum concentrations of progesterone and cortisol were positively correlated (p<0.01) in a small group of captive female African elephants (ref 98). Importantly, in addition to age, sex, reproductive status, handling method, and location, the discussion should include thoughts on how diurnal fluctuations in salivary cortisol levels may have impacted findings – especially given the different collection time in Kronberg.

L798-813: We included the discussion on the potential impact of the diurnal variation of cortisol secretion on the findings in section 4.4 (see also the reviewer’s previous comment).

We included reproductive status and ovarian cycle phase as other factors that may influence salivary cortisol levels in the present study.

L152-154: We also list reproductive status and ovarian cycle phase as factors that contribute to individual variation.

L239 (Table 1): We state reproductive status and ovarian cycle phase of females.

L460-465: We describe the analysis of the effect of reproductive status and ovarian cycle phase. We analyzed the effect of these factors on the individual level (see Glaeser et al. 2020). We also fitted both factors as another fixed effect in the LMMs described in Table 2 where the effect was not significant. We can also provide this analysis if required.

L575-602: We present the results of the effect of reproductive status and ovarian cycle phase including two new figures.

L693-719: We discuss the findings on the effect of reproductive status and ovarian cycle phase including two new figures.

While the paper is generally well-written, a few sentences would benefit from some grammatical revisions – e.g., lines 73-74: change “In zoo animals, by contrast, it can be trained as part of daily routines” to “Zoo animals, by contrast, can be trained for sample collections as part of daily routines”.

L99-100: We have reworded this sentence and paid attention to correct grammar.

References

Hambrecht, S.; Oerke, A.-K.; Heistermann, M.; Dierkes, P.W. Diurnal variation of salivary cortisol in captive African elephants (Loxodonta africana) under routine management conditions and in relation to a translocation event. Zoo Biol. 2020, 39, 186–196, doi:10.1002/zoo.21537.

Glaeser, S.S.; Edwards, K.L.; Wielebnowski, N.; Brown, J.L. Effects of physiological changes and social life events on adrenal glucocorticoid activity in female zoo-housed Asian elephants (Elephas maximus). PLOS ONE 2020, 15, e0241910, doi:10.1371/journal.pone.0241910.

Round 2

Reviewer 1 Report

Dear authors, 

Thank you for taking the time to make those modifications and answer all of my questions. I would be more than happy to support publication of this paper.